# Deoxynivalenol Induces Inflammatory Injury in IPEC-J2 Cells via NF-κB Signaling Pathway

**DOI:** 10.3390/toxins11120733

**Published:** 2019-12-16

**Authors:** Xichun Wang, Yafei Zhang, Jie Zhao, Li Cao, Lei Zhu, Yingying Huang, Xiaofang Chen, Sajid Ur Rahman, Shibin Feng, Yu Li, Jinjie Wu

**Affiliations:** Department of Veterinary Medicine, College of Animal Science and Technology, Anhui Agricultural University, Hefei 230036, China; wangxichun@ahau.edu.cn (X.W.); djdshhz@163.com (Y.Z.); zhaojie866@ahau.edu.cn (J.Z.); caoli@ahau.edu.cn (L.C.); zhuleizl@ahau.edu.cn (L.Z.); huangyingying@ahau.edu.cn (Y.H.); chenxiaofang0912@163.com (X.C.); dr_sajid226@yahoo.com (S.U.R.); fengshibin@ahau.edu.cn (S.F.)

**Keywords:** deoxynivalenol, IPEC-J2, cell damage, NF-κB inflammatory signal pathway

## Abstract

The aim of this study was to investigate the effects of deoxynivalenol (DON) exposure on the inflammatory injury nuclear factor kappa-B (NF-κB) pathway in intestinal epithelial cells (IPEC-J2 cells) of pig. The different concentrations of DON (0, 125, 250, 500, 1000, 2000 ng/mL) were added to the culture solution for treatment. The NF-κB pathway inhibitor pyrrolidine dithiocarbamate (PDTC) was used as a reference. The results showed that when the DON concentration increased, the cell density decreased and seemed damaged. With the increase of DON concentration in the culture medium, the action of diamine oxidase (DAO) in the culture supernatant also increased. The activities of IL-6, TNF-α, and NO in the cells were increased with the increasing DON concentration. The relative mRNA expression of *IL-1β* and *IL-6* were increased in the cells. The mRNA relative expression of *NF-κB p65*, *IKKα,* and *IKKβ* were upregulated with the increasing of DON concentration, while the relative expression of *IκB-α* mRNA was downregulated. At the same time, the expression of NF-κB p65 protein increased gradually in the cytoplasm and nucleus with a higher concentration of DON. These results showed that DON could change the morphology of IPEC-J2 cells, destroy its submicroscopic structure, and enhance the permeability of cell membrane, as well as upregulate the transcription of some inflammatory factors and change the expression of NF-κB-related gene or protein in cells.

## 1. Introduction

Mycotoxins are widely found in human and animal foods. Deoxynivalenol (DON) is mainly produced by *Fusarium graminearum* which is prone to crops such as corn and wheat [1,2,3]. China is widely contaminated by such kind of mycotoxins, especially in the area of Yangtze and Yellow rivers. In different grains the climate is favorable to the growth of mold and their relevant toxin production [4]. Previously, from 2016–2017, 827 complete feed samples of pigs and 724 components of feed were chosen from various provinces of China for analysis, which shows that the last standard rate of DON ingredient was above 74.5%, and 450.0–4381.5 μg/kg was the average concentration [5]. Hence, further studies on DON become even highly significant.

It has been reported that DON can cause animals to food refusal, organ damage, and increase the risk of disease [6,7,8]. It is generally believed that DON exerts many toxic effects such as neurotoxicity, immuno-toxicity, intestinal toxicity, and cytotoxicity [9,10]. Pigs are the utmost susceptible animal to DON [11]. When DON enters the animal’s body, it first acts on the digestive tract. When the animal ingests food or feed containing DON, the intestinal epithelial cells will be exposed to a high concentration of DON, which can potentially affect the intestinal function of the animal [12,13,14].

Small intestinal epithelial cells can effectively inhibit the propagation of potentially harmful micro-organisms between the inner and lower mucosa [15]. Infection mechanisms of intestinal epithelial cells include bacterial attachment or invasion of cells, proliferation of pathogens, and host cell responses [16]. Experiments have shown that the use of 2000 ng/mL DON to treat pig intestinal epithelial cells (IPEC-J1 and IPEC-J2), decreased ZO-1 protein expression in cells, indicating that DON can make the mechanical integrity of intestinal barrier compromised [17]. At the same time, the transportation and absorption of nutrients in epithelial cells have more impact, through which the substances needed for the growth and development of the body are provided [18]. Maintaining the homeostasis of the intestinal tract, at this point, the structure and function of epithelial cells and lymphocytes in the intestinal mucosa play a key role. For example, lymphocytes in the mucosa can be released by releasing a series of antibody molecules to neutralize toxins [19]. Simultaneously, there is a chemical barrier in the intestines, which is composed of various liquids secreted by epithelial cells mixed with bacteriostatic peptide secreted by the microflora in the intestinal tract to cover the mucus on the surface of epithelial cells [20]. Therefore, the epithelial cells are also considered to be the medium of an early natural immune response [21].

IPEC-J2 is isolated from the jejunal epithelial tissue of newborn piglets and is a small intestine cell model of piglets cultured in vitro [22]. In view of the high homology of the intestinal structure and function of pigs and humans, studies conducted by IPEC-J2 cells can provide a theoretical basis for human medicine [22]. The pig intestinal epithelial cell line IPEC-J2 cells mostly form a cell monolayer when cultured in vitro, and occasionally there are stratified regions. The monolayer consists of cuboid cells interspersed with flat cells. No goblet cells were observed [16]. During cell injury or growth NF-κB plays an important role by acting as a regulatory pathway [23], the classical NF-κB pathway affecting numerous functions of cells that damage cells by enhancing inflammatory related genes expression, growth, and immunity [24].

In this study, we increased the concentration of DON in the culture medium, and selected the NF-κB pathway inhibitor PDTC as a reference, aimed to explore the phenomenon by which DON persuades inflammatory impairment in IPEC-J2 cells via NF-κB signaling pathway by the features of morphological structure, cell viability, cellular inflammatory mediators, and expression of pathway-related gene proteins, etc.

## 2. Results

### 2.1. Effects of DON on Cell Viability Rate 

As can be seen from Figure 1, the viability rate of IPEC-J2 cells decreased with the increasing concentration of DON in the cell culture medium. Compared with the control cells, when the DON concentration reached 250 ng/mL the viability rate of cells in each treatment group was significantly decreased (*p* < 0.05), and further increased to 500 ng/mL and above (*p* < 0.01).

### 2.2. Effects of DON on Morphological Changes in Intestinal Epithelial Cells

As shown in Figure 2, the cells are in an irregular shape and closely connected when growing normally with a large number of cells (Figure 2A). Subsequently, (125, 250, 500, 1000, and 2000 ng/mL) concentrations of DON were applied to the cells, the cell density steadily reduced, and when the DON concentration enhance in the culture medium, the cells indicate irregular architectures (Figure 2B–F).

### 2.3. Effect of DON on the Submicroscopic Structure of Cells

Figure 3 shows that, the morphology of the cells in the control group is intact, the organelles in the cytoplasm are normal, and the distribution of chromatin in the nucleus is relatively uniform (Figure 3A1,A2). When the cells were exposed to 125 ng/mL of DON (Figure 3B1,B2), there was no obvious pathological change seen in the nucleus, but rupture of mitochondrial mites seen in the cytoplasm while the number of cristae was also reduced. With the increasing concentration of DON in the culture solution (Figure 3C–E), the ribosome in the cytoplasm gradually decreased, and the organelles such as mitochondria and endoplasmic reticulum gradually vacuolized and, the chromatin marginal aggregation in the nucleus increased. When the concentration reached 2000 ng/mL (Figure 3F1,F2), the chromatin in the nucleus aggregated and disappeared, and the organelle vacuolization becomes severe.

### 2.4. Effect of DON on DAO Activity in Cell Culture Supernatant

As shown in Figure 4, during comparison with the control group, increasing the concentration of DON in the culture, the amount of DAO was expressively enhanced in the culture supernatant of IPEC-J2 cells (*p* < 0.01). The DAO content in the supernatant of the experimental group was significantly decreased (*p* < 0.01), when the DON was added with PDTC.

### 2.5. Effect of DON on the Activity of Inflammatory Factors in Cells

The IL-6, TNF-α, and NO levels in the cells were increased with a higher concentration of DON when matched to the control group, while PDTC reduces the content of these inflammatory mediators presented in Figure 5. Compared to the control group, when the concentration of DON in the culture medium reached 125 ng/mL and above, the concentration of IL-6, TNF-α, and NO in the cells were significantly increased (*p* < 0.01). Compared with the 1000 ng/mL DON group without PDTC, the group after adding PDTC significantly reduced the contents of TNF-α and NO in the cells (*p* < 0.01).

### 2.6. DON Effect on the mRNA Expression of Inflammation-Related Genes in Cells

The data in Figure 6 illustrated that DON increased the relative expression of inflammatory mediators in IPEC-J2 cells, and decreased after the addition of pathway inhibitors. Among them, the relative mRNA expression of *IL-1β* in the cells increased with the increasing of DON concentration, and was significantly higher than the control group when the DON concentration was 250 ng/mL and above (*p* < 0.01). The relative expression of *IL-1β* in the test group of 1000 ng/mL DON plus PDTC was significantly lowered than that in the test group of 1000 ng/mL DON without PDTC (*p* < 0.05). In addition, the relative expression of *IL-6* gene in the cells increased significantly with the increasing of DON concentration (*p* < 0.01).

In this experiment, with the increasing of DON contents in the culture medium, the relative expression levels of *NF-κB p65*, *IKKα*, *IKKβ*, *iNOS,* and *COX-2* mRNA in the cells were upregulated to different degrees compared with the control group. Compared with the unadded test group, the relative expression of some genes in the PDTC test group was downregulated by different degrees. When the concentration of DON in the culture medium was 125 ng/mL and above, the mRNA expression levels of *IKKα*, *IKKβ*, *iNOS,* and *COX-2* were significantly upregulated compared with the control group (*p* < 0.01). After adding PDTC to the culture medium containing 1000 ng/mL DON, the relative expression of *iNOS* mRNA gene in the cells was significantly decreased (*p* < 0.01). Compared to the control group, when the concentration of DON in culture medium increased to 250 ng/mL, the relative expression of *NF-κB p65* mRNA significantly upregulated (*p* < 0.05), and the relative expression of *IκB-α* mRNA downregulated (*p* < 0.05). When the concentration of DON increased to 500 ng/mL and above, the relative expression of *NF-κB p65* mRNA significantly upregulated compared with the control group (*p* < 0.01) and the relative expression of *IκB-α* mRNA gene was a significant downregulation (*p* < 0.01). After the addition of pathway inhibitors in the experimental group, the relative expression of *NF-κB p65* gene was significantly downregulated (*p* < 0.01), and the relative expression of *IκB-α* mRNA gene was significantly upregulated (*p* < 0.05).

### 2.7. Effect of DON on the Expression of NF-κB p65 Protein in the Nucleus

In this study, the nuclear expression of NF-κB p65 was detected by EMSA. As shown in Figure 7, with the increasing concentration of DON, the nuclear expression of NF-κB p65 was increased, as well as significantly (*p* < 0.01) increased in DON treatment group, while the expression of NF-κB p65 protein in the 10 μM PDTC-treated cells decreased.

### 2.8. Effect of DON on the Expression and Distribution of NF-κB p65 Protein in Cells

When the concentration of DON in the culture medium increased, the green fluorescence intensity of NF-κB p65 protein in the cell, especially in the nucleus also increased gradually as shown in Figure 8, indicating that DON can enhance the expression of NF-κB p65 and nuclear expression.

## 3. Discussion

Many studies have found that zearalenone (ZEA) and DON can reduce the survival rate of IPEC-J2 cells with a certain time and dose dependence manner [13,25]. Research has shown that DON concentrations at 250 and 1000 ng/mL can significantly reduce cell counts in a dose-dependent manner. When the DON concentration is 1000 ng/mL, it caused cell damage, including cell monolayer autolysis, and cell loss [26]. According to the results of this experiment, when the DON concentration in the culture medium was 125 ng/mL, it shows little effect on cell viability. When the concentration increased to 250 ng/mL, the cell survival rate was significantly lower than that of the control group. The cell survival rate decreased significantly, if the culture medium was 250 ng/mL or higher concentration.

Intestinal epithelial cells are not only selectively permeable as a nutrient-absorbing filter, but are also considered to be the first line of defense against foreign antigens, such as pathogens and toxins from the intestinal lumen [27]. DAO is an intracellular enzyme in mammalian intestinal epithelial cells that put forth a protective effect on the intestinal mucosa. When small intestinal mucosal cells are necrotic, DAO in cells will be released [28]. Thus, the extracellular fluid DAO content can reflect the destruction of cells in small intestine. In this experiment, with the higher concentration of DON in the culture solution, the DAO content in the culture supernatant also increased significantly, indicating that DON can damage IPEC-J2 cells and enhance the permeability of the cell membrane [29].

DON influencing intestinal barrier integrity and induced pro-inflammatory cytokines abnormal expression [30]. Previously it was found that DON can cause an increase in the activity of cytokines such as IL-1β, IL-6, and TNF-α in IPEC-J2 cells [31]. In addition, DON also upregulates the activity of TNF-α in IECs cell [30]. According to the test results of the present experiment, the activity of inflammatory mediators in IPEC-J2 cells and the relative expression of *IL-1β* and *IL-6* mRNA in the cells increased with the increasing of DON concentration in the culture medium, while PDTC reduced these trends. This indicated that DON cause an inflammatory response in IPEC-J2 cells and might affect the normal physiological state of the cells via the NF-κB pathway.

Previously, it commonly exists in the shape of a p50/65-IκBα dimer [32]. Different studies showed that when active NF-κB enhanced in a dose-dependent pattern, it initiates to be effective [29,33]. Furthermore, NF-κB pathway regulates COX-2 activation and additional transcription factors [34,35], which demonstrates that this pathway plays a significant role in the inflammatory impairment of the intestine.

In this experiment, the intracellular NO activity and the relative mRNA expression of *iNOS* increased with the dosage of DON, and the relative expression of *COX-2* gene, which is the same as the inducible enzyme, also increased in cells. These results indicate that IPEC-J2 cells are stimulated by DON to produce a corresponding inflammatory response. On the other hand, the NF-κB pathway is primarily thought to be a potential pathogenic factor that exerts deleterious effect on cells when it is excessively or inappropriately activated [24,36]. In our results, the relative mRNA expression of *NF-κB p65*, *IKKα,* and *IKKβ* genes were upregulated, while the mRNA relative expression of *IκB-α* was downregulated to varying degrees. At the same time, the inhibitory changes were inhibited in different degrees.

## 4. Conclusions

The current study indicated that DON alters IPEC-J2 cells morphology, destroys the submicroscopic structure, boosts cell membrane permeability, and upregulates the transcription of various associated inflammatory factors in cells and alters the expression of NF-κB-related gene or protein in cells. We also concluded that the distribution and content of NF-κB p65 in the intracellular and nucleus further indicated that DON induced inflammatory damage of IPEC-J2 cells through the NF-κB signaling pathway.

## 5. Materials and Methods

### 5.1. Chemical and Reagents

DON (CAS No. D0156-5MG) was procured from Sigma (Sigma Chemical Co. St. Louis, MO, USA). Porcine IPEC-J2 cells were obtained from a cell bank in Wuhan academy of agricultural sciences, Wuhan, China. RPMI 1640, SuperScript III kit and Sybr qPCR mix were bought from Thermo Fisher Scientific, Waltham, MA, USA. Fetal bovine serum (FBS) was procured from Clark Bioscience, Richmond, VA, USA. NF-κB inhibitor (Pyrrolidine dithiocarbamate, PDTC) was procured from Beyotime Biotechnology, Shanghai, China. Cell counting kit-8 (CCK-8) kits were obtained from Dojindo Laboratories, Tokyo, Japan. The ELISA kit was bought from Senbeijia Biological Technology, Nanjing, China. BSA was bought from biosharp Company, Beijing, China. The primary NF-κB p65 polyclonal antibody (Product number: 10745-1-AP) was obtained from Proteintech Group, Inc, Rosemont, IL, USA. FITC-goat anti-rabbit IgG antibody (Product number: BA1105) was purchased from Boster Company, Wuhan, China. Trizol reagent was purchased from Invitrogen Biotechnology Co., Ltd., Shanghai, China.

### 5.2. Cell Culture and Treatments

The intestinal epithelial cells (porcine IPEC-J2) were cultivated in culture bottles (4 × 6 cm) in RPMI 1640 added with 10% (*v*/*v*) FBS, 100 U/mL penicillin, and 100 μg/mL streptomycin, and cultured at 37 °C and a moistened incubator with 5% CO_2_. DON solution of 1 mg/mL stock was prepared by liquefying 1 mg DON in 1 mL RPMI 1640 comprising FBS of 10% (*v*/*v*). In this experiment diluted concentrations of 125, 250, 500, 1000, and 2000 ng/mL DON were used [37].

IPEC-J2 cells in logarithmic growing phase (1 × 10^5^ cells/mL) were cultivated in 96-well plates in 100 μL RPMI 1640 for 24 h; and were cured with (0, 125, 250, 500, 1000, and 2000 ng/mL) various DON concentrations for 24 h for the evaluation of cell viability assay. For the estimation of NF-κB pathway in response to DON acquaintance, PDTC, the NF-κB inhibitor was added to the two experimental groups (0, 1000 ng/mL DON) 30 min earlier than the treatment with DON. The supernatants of cell culture were gathered to know DAO releasing. The collected cells were observed for morphological, inflammatory mediator activity, and studies of NF-κB pathway-associated gene or proteins [37].

### 5.3. IPEC-J2 Cell Morphology by Optical Microscope

The IPEC-J2 cells in the logarithmic growth phase were taken, and the cell density was adjusted to 1 × 10^5^ per mL in a 6-well cell culture plate. After culturing to adherence, the culture solution was changed to a cell culture medium containing different concentrations of DON, and after 24 h, the cell growth of each group was observed under an inverted light microscope (Chongguang Industry Co., Ltd. Chongqing, China).

### 5.4. IPEC-J2 Cell Morphology by Transmission Electron Microscopy (TEM)

Cells were collected at the bottom of the centrifuge tube and 2.5% glutaraldehyde were used to be fixed in for 4 h, dehydrated, soaked, embedded, ultrathin sections, lead citrate stained, and washed. The cells ultrastructure was detected via a high resolution transmission electron microscope TEOL-2010 (Electronics Corporation, Tokyo, Japan).

### 5.5. Detection of Cell Viability

IPEC-J2 cells in the logarithmic growth phase were seeded in 96-well plates at about 1000 cells per well, cultured until the cells were attached, and DON was used. After 24 h, 10 μL of CCK-8 reagent was added to each well and cultured for 2 h. The cell viability was calculated by using the absorbance at a wavelength of 450 nm.

### 5.6. Detection of Inflammatory Mediators and Intestinal Permeability Indicators

The cells were treated with DON and PDTC, and then cultured in an incubator for 24 h. The culture solution was centrifuged at 1000 rpmmin^−1^ for 5 min to obtain a cell culture supernatant. The cells were collected in a cell via a subculture method, and the cell lysate was collected. The NO, IL-6, and TNF-α in the cell lysate were used according to the method of the ELISA kit (Senbeijia Biological Technology, Nanjing, China). The activity and the DAO in the cell culture supernatant was measured, and its activity was calculated according to a self-drawn standard curve.

### 5.7. Quantitative Real-Time PCR

As per the manufacturer’s protocol, total RNA was isolated from cells via a Trizol reagent. Nanodrop lite (Thermo Inc, Waltham, MA, USA) was used to examine the concentrations of RNA. The reverse transcription was accomplished via Super-Script III First-strand cDNA Synthesis Mix (Thermo Inc, USA). Real-time PCR was performed with SybrGreen qPCR Mastermix (Thermo Inc, USA). The overall samples were assayed three times. The reaction mixtures were incubated in a 7900 fast real-time PCR system (Applied Biosystems, Foster City, CA, USA)). The program comprised of 1 cycle at 95 °C for 120 s, 40 cycles at 94 °C for 20 s, 60 °C for 20 s, and 72 °C for 30 s. The gene relative expression levels were calculated according to the 2^−ΔΔCT^ method. In real-time PCR analysis, β-actin was used as a housekeeping gene to estimate levels of mRNA for normalization. The primer sequences were synthesized by Sangon Biotech Co., Ltd. (Shanghai, China) and described in Table 1.

### 5.8. Immunofluorescence 

Phosphoryl-NF-κB p65 localization was quantified via an immunofluorescence technique. Paraformaldehyde (*v*/*v*, 1/25) was used for IPEC-J2 cells fixation for 30 min at a temperature of 37 °C. After a PBS wash (0.1 mM, pH7.4), permeabilised in 0.5% Triton (Triton×100, Sigma, Harz Lower Saxony, Germany) for 20 min, and for 20 min blocked with 5% BSA, as well as hatched with the anti-phosphoryl-NF-κB p65 antibody (diluted 1:100) for the whole night at a temperature of 4 °C. After washing with PBS (0.1 mM, pH7.4) for the second time, the secondary antibody was used to incubate the cells for 1 h at room temperature. Coverslips were washed two times via PBS (0.1 mM, pH7.4), and hatched with the goat anti-rabbit IgG antibody for 1 h in the absence of light, and hatched in a DAPI staining solution for 10 min. After that it was washed again in PBS. The fluorescence was monitored using an Olympus-fluoview ver.3.1 viewer (Olympus Corporation, Miyazaki Prefecture, Kyushu, Japan) [38].

### 5.9. Electrophoretic Mobility Shift Assays (EMSAs)

NF-κB DNA-binding activity was examined by EMSA. The cytoplasmic and nuclear protein extraction kit (Jiangsu KeyGEN BioTECH Corp., Ltd., Nanjing, China) was used to prepare nuclear extract. The consensus nucleotide sequence for NF-κB was 5’-AGT TGA GGG GAC TTT CCC AGG C-3’. The EMSA binding reaction was performed by the EMSA kit (Jiangsu KeyGEN BioTECH Corp., Ltd.). A nuclear extract was incubated in a 5× binding reaction buffer containing the biotinylated probe. After incubated at room temperature for 20 min, the reaction mixture was electrophoresed on a non-denaturing 6.5% polyacrylamide gel and then transferred to a nylon membrane. The shifted mixture and the membrane were UV-cross-linked and the ECL kit used to detect was obtained from (Jiangsu KeyGEN BioTECH Corp., Ltd.). For the super shift assay, 1 μg of antibody against NF-κB p65 was added together with the nuclear extract.

### 5.10. Statistics Analysis

For the calculation of protein expression of average optical density (OD), the Image-Proplus 6.0 Analysis Software (Media Cybernetics, Shanghai, China) were used. The obtained data were presented as means ± SD (n = 10). Statistical analysis was performed by the Statistical Program for Social Sciences (SPSS) software version 19.0 (IBM Corporation, Armonk, NY, USA). Analysis of variance (ANOVA) was performed for the multiple comparison of different groups. The histogram was designed by using the software of Graph Pad Prism version 5.0 (San Diego, California, CA, USA).

## Figures and Tables

**Figure 1 toxins-11-00733-f001:**
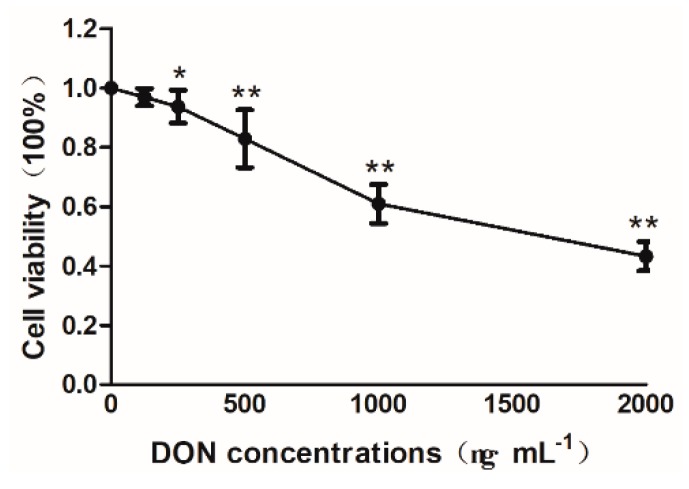
Effects of DON on cell viability rate in cells. The concentrations are 500, 1000, 1500, and 2000 in order. The above data presented as means ± SD of three independent tests (n = 10). Note: * indicates a significant difference compared with the data of the control group (*p* < 0.05), and ** indicates highly significant compared with the control group (*p* < 0.01).

**Figure 2 toxins-11-00733-f002:**
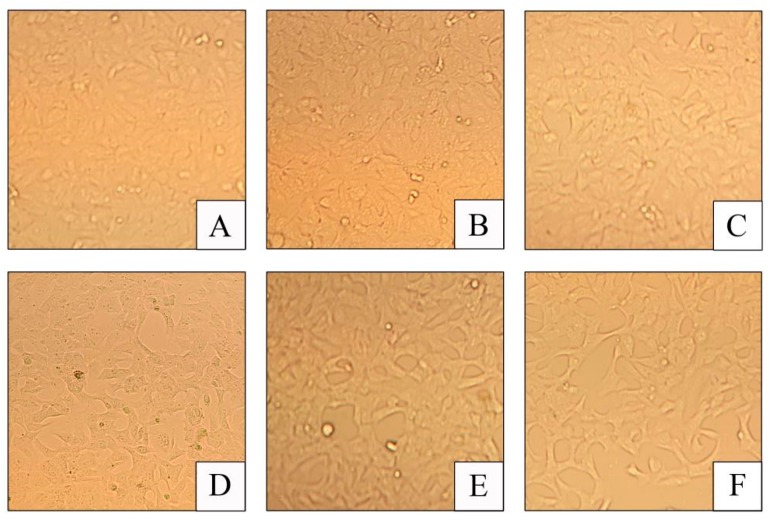
Effect of DON on growth state in cells (400×). (**A**): Control group; (**B**): 125 ng/mL deoxynivalenol (DON) group; (**C**): 250 ng/mL DON group, (**D**): 500 ng/mL DON group, (**E**): 1000 ng/mL DON group, and (**F**): 2000 ng/mL DON group.

**Figure 3 toxins-11-00733-f003:**
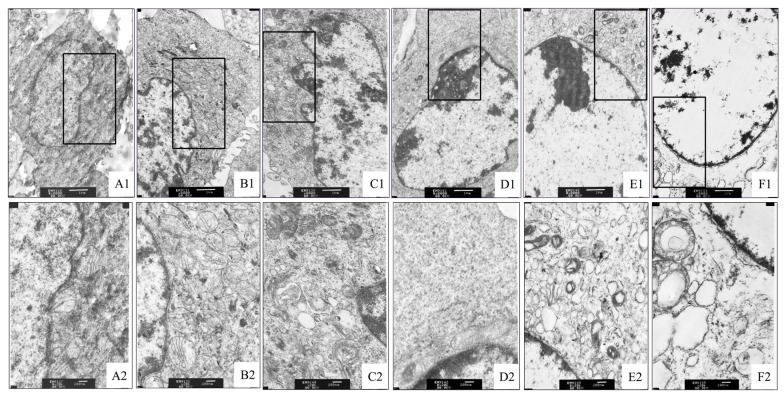
Effect of DON on the ultrastructure of cells. (**A1**,**A2**: Control group), (**B1**,**B2**: 125 ng/mL deoxynivalenol (DON) group), (**C1**,**C2**: 250 ng/mL DON group), (**D1**,**D2**: 500 ng/mL DON group), (**E1**,**E2**: 1000 ng/mL DON group), (**F1**,**F2**: 2000 ng/mL DON group). Image amplification of **A1**–**F1** is 6000×; image amplification of **A2**–**F2** is 20,000×. The black frame in A1, B1, C1, D1, E1, and F1, are enlarged to A2, B2, C2, D2, E2, and F2 respectively.

**Figure 4 toxins-11-00733-f004:**
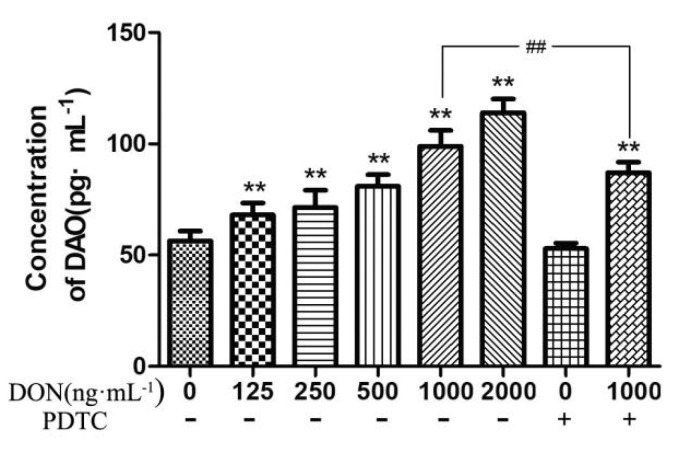
Effects of DON on the activity of DAO in cell culture fluid. The above data shows as means ± SD of three independent tests (n = 10). Note: ** shows extremely significant difference (*p* < 0.01), compared with the control group. ## indicates highly significant (*p* < 0.05) compared with the test group and with the same DON concentration but no addition of PDTC. These rules are also applied for Figure 5, Figure 6 and Figure 7.

**Figure 5 toxins-11-00733-f005:**
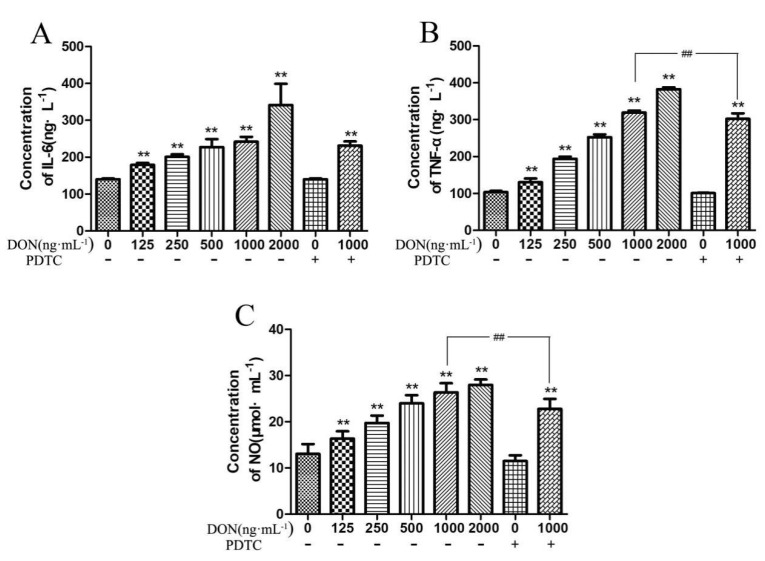
Effect of DON on the concentration of inflammatory factors in cells. (**A**–**C**) IL-6, TNF-α, and IL-1β expression. The collected data are shown as means ± SD of three independent tests (n = 10). Note: ** shows extremely significant difference (*p* < 0.01), compared with the control group. ## indicates highly significant (*p* < 0.05) compared with the test group and with the same DON concentration but no addition of PDTC.

**Figure 6 toxins-11-00733-f006:**
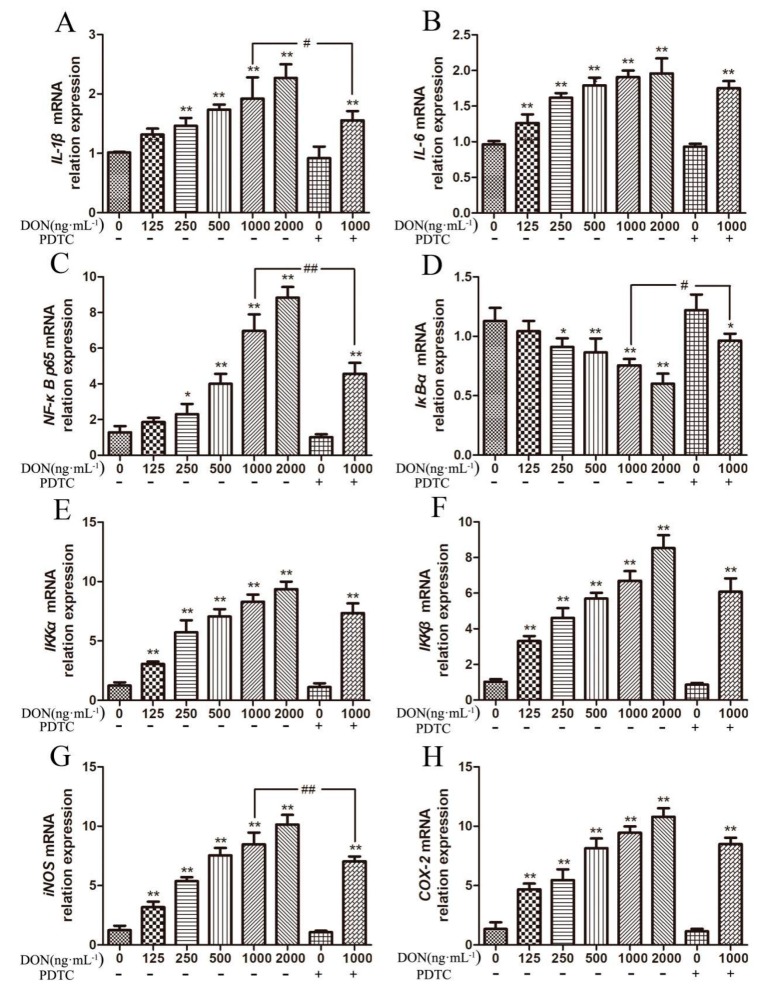
DON influence on the mRNA expression of NF-κB pathway associated genes in cells. (**A**–**H**) the expression of *IL-1β, IL-6*, *NF-κB p65*, *IκB-α*, IKKα, *IKKβ*, *iNOS,* and *COX-2* mRNA. Note: * designates a significant change as compared to the control group (*p* < 0.05), and ** shows extremely significant difference (*p* < 0.01), compared with the control group. # Indicates significant (*p* < 0.05) compared with the test group and with the same DON concentration but no addition of DON. ## Indicates highly significant (*p* < 0.01) compared with the test group and with the same DON concentration but no addition of PDTC.

**Figure 7 toxins-11-00733-f007:**
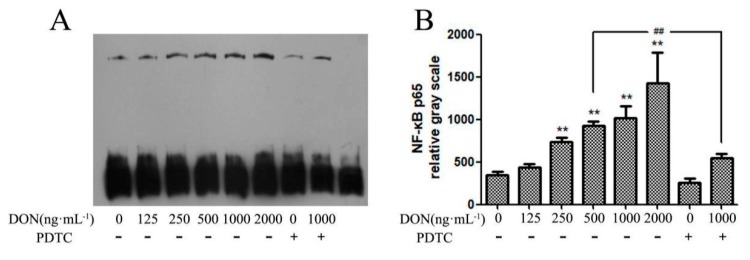
Effect of DON on the transcriptional activity of NF-κB p65 in cells. (**A**) Western blotting showing NF-κB p65. (**B**) Impact of DON on the protein expression of NF-κB p65. Note: ** shows extremely significant difference (*p* < 0.01), compared with the control group. ## indicates highly significant (*p* < 0.05) compared with the test group and with the same DON concentration but no addition of PDTC.

**Figure 8 toxins-11-00733-f008:**
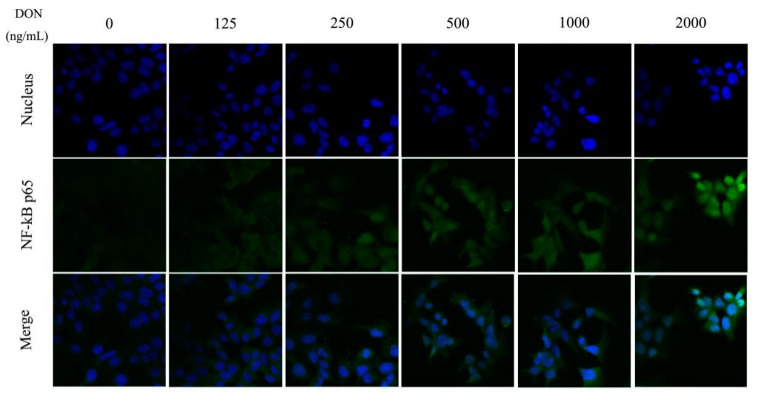
Effect of DON on the nuclear expression of NF-κB p65 protein in cells (800×).

**Table 1 toxins-11-00733-t001:** Parameters of primer for inflammatory cytokines and β-actin genes.

Genes	Accession Number	Primers	Sequences (5′–3′)	Production Size/bp
*β-actin*	AY_550069.1	Forward	AGATCAAGATCATCGCGCCT	170
		Reverse	ATGCAACTAACAGTCCGCCT	
*IL-1β*	NW_018085011.1	Forward	TCAGCACCTCTCAAGCAGAA	120
		Reverse	GACCCTCTGGGTATGGCTTT	
*IL-6*	NC_010451.4	Forward	CTGCAGTCACAGAACGAGTG	131
		Reverse	GACGGCATCAATCTCAGGTG	
*NF-κB p65*	NM_001114281.1	Forward	GGGGCGATGAGATCTTCCTG	110
		Reverse	CACGTCGGCTTGTGAAAAGG	
*IκB-α*	NC_010449.5	Forward	GGAGTACGAGCAGATGGTGA	157
		Reverse	TTCCATGGTCAGTGCCTTCT	
*iNOS*	NC_010454.4	Forward	GGGTCAGAGCTACCATCCTC	114
		Reverse	CGTCCATGCAGAGAACCTTG	
*IKKα*	NC_010456.5	Forward	CACTCTTACAGCGACAGCAC	145
		Reverse	CCACCTTGGGCAGTAGATCA	
*IKKβ*	NT_176339.1	Forward	ACCTGGCTCCCAACGACTT	184
		Reverse	AGATCCCGATGGATGATTCTG	
*COX-2*	NC_010451.4	Forward	TGCGGGAACATAATAGAG	90
		Reverse	GTATCAGCCTGCTCGTCT	

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
