# Peer review of "Deoxynivalenol Induces Inflammatory Injury in IPEC-J2 Cells via NF-κB Signaling Pathway"

_toxins, 2019, doi:10.3390/toxins11120733_

Round 1

Reviewer 1 Report

The mechanism of porcine mucosal injury related to deoxynivalenol (DON) exposure is still not fully characterized which prompted the authors of the manuscript to conduct research on the mechanism by which DON induces inflammatory damage in  IPEC-J2 cells via NF-κB signaling pathway. The investigated IPEC-J2 cells are intestinal porcine enterocytes isolated from the jejunum of a neonatal unsuckled piglet and they mimic the human physiology more closely than any other cell line of non-human origin. Therefore, obtained results can be valuable for human medicine. The IPEC-J2 cells were treated by different concentrations of DON (0, 125, 250, 500, 1000, and 2000 ng/mL). The effects of DON on cell viability rate, morphological changes and submicroscopic structure, diamine oxidase activity (DAO) in cell culture supernatant, activity of inflammatory factors and mRNA expression of inflammation-related genes (NF-κB p65, IKKα, IKKβ, iNOS and COX-2 mRNA) and on the expression of NF-κB p65 protein in the nucleus of studied in IPEC-J2 cells were evaluated. The results obtained indicate changes in the morphology of IPEC-J2 cells under the exposure of DON. The toxin caused destruction of the submicroscopic structure of the cells, and enhanced the permeability of cell membrane. Up-regulation of the transcription of some inflammatory factors in cells and change the expression of NF-κB-related gene or protein in cells were noticed.

The manuscript is written in a good manner however requires some minor corrections before acceptance.

LIST OF SUGGESTED CHANGES

ABSTRACT

Page 1, line 9:  Please enter the full name for DAO - abbreviation used for the first time

INTRODUCTION

General remark: The introduction is a bit unfocused on the topic of work indicated in the title. The NF-κB pathway should be briefly discussed.

Page 1, line 25: the Latin name of the mold should be in italics.

Page 1, line 43: There is no spaces between 2000 ng · mL-1 and the DON abbreviation

Page 2, lines 60 – 65: The section: “In this study, we chose to increase the concentration of DON in the culture medium, and selected the NF-κB pathway inhibitor PDTC as a reference. It is hoped that the morphological structure, cell viability, cellular inflammatory mediators, and expression of pathway-related gene proteins can be used. Further studies were conducted on the mechanism by which DON induces inflammatory damage in  IPEC-J2 cells via NF-κB signaling pathway.” - is incomprehensible, especially:

To increase the concentration of DON –from which to which level. Selected the NF-κB pathway inhibitor PDTC as a reference - reference to what? It is hoped that the morphological structure, cell viability, cellular inflammatory mediators, and expression of pathway-related gene proteins can be used – can be used for what purpose? Further studies were conducted on the mechanism by which DON induces inflammatory damage in IPEC-J2 cells via NF-κB signaling pathway - from the subject of the work it follows that this was the main part of the work not further

In cited section the aim of the study should be clearly presented.

DISCUSSION

Page 8, line 177: Thre is no spaces between inreasing and concentration

MATERIALS AND METHODS

Page 9, line 212: correct the spelling (culture not culturet)

Page 9, line 229: please harmonize the spelling of the milliliter unit (mL not ml)

FIGURES

Figure 1

The title: Effects of DON on cell viability rate in cells - is incomprehensible (cell viability rate in cells?) Are the results on the y-axis really expressed in%?

Author Response

Point 1: Page 1, line 9: Please enter the full name for DAO - abbreviation used for the first time

Response 1: Thank you for your valuable suggestions. We have entered the full name of DAO according to the reviewer's suggestion in our revised manuscript.

Point 2: General remark: The introduction is a bit unfocused on the topic of work indicated in the title. The NF-κB pathway should be briefly discussed.

Response 2: Thank you for your valuable comment. We have read some more articles about NF-κB signaling pathway and made a brief discussion in introduction.

Point 3: Page 1, line 25: the Latin name of the mold should be in italics.

Response 3: We have changed fungus to italics. Thank you for your careful review.

Point 4: Page 1, line 43: There is no space between 2000 ng · mL-1 and the DON abbreviation

Response 4: Thank you for your careful review. We have added spaces between 2000 ng · mL-1 and the DON abbreviation and checked the entire document for these kinds of errors.

Point 5: Page 2, lines 60 – 65: The section: “In this study, we chose to increase the concentration of DON in the culture medium, and selected the NF-κB pathway inhibitor PDTC as a reference. It is hoped that the morphological structure, cell viability, cellular inflammatory mediators, and expression of pathway-related gene proteins can be used. Further studies were conducted on the mechanism by which DON induces inflammatory damage in IPEC-J2 cells via NF-κB signaling pathway.” - is incomprehensible, especially:

To increase the concentration of DON –from which to which level. Selected the NF-κB pathway inhibitor PDTC as a reference - reference to what? It is hoped that the morphological structure, cell viability, cellular inflammatory mediators, and expression of pathway-related gene proteins can be used – can be used for what purpose? Further studies were conducted on the mechanism by which DON induces inflammatory damage in IPEC-J2 cells via NF-κB signaling pathway - from the subject of the work it follows that this was the main part of the work not further

In cited section the aim of the study should be clearly presented.

Response 5: Thank you for your valuable comment and important point. We have set the concentration of DON as 0-2000 ng·mL-1, 0, 125, 250, 500, 1000 and 2000 ng·mL-1 respectively. We selected PDTC as the inhibitor of NF-κB pathway, when the inhibitor PDTC was added, the response of this pathway was blocked, and the damage in IPEC-J2 cells was reduced when the DON added with PDTC. It might be concluded that the damage was via NF-κB signaling pathway. According to your valuable suggestions, we have specified the main content of this experiment in the introduction of this manuscript

Point 6: Page 8, line 177: There are no spaces between increasing and concentration

Response 6: Thank you for your comment. We have added spaces between increasing and concentration.

Point 7: Page 9, line 212: correct the spelling (culture not culture)

Response 7: Thank you for your careful review. We have revised the spelling of culture.

Point 8: Page 9, line 229: please harmonize the spelling of the milliliter unit (mL not ml)

Response 8: Thank you for your valuable suggestion. We have harmonized the spelling of mL according to the reviewer's suggestion.

Point 9: Figure 1-The title: Effects of DON on cell viability rate in cells - is incomprehensible (cell viability rate in cells?) Are the results on the y-axis really expressed in%?

Response 9: We have revised the title name in our manuscript. Cell viability rate represent the percentage of living cells. Therefore, the y-axis unit is %, and the number of 1 represents 100% in y-axis. Thank you for your valuable comment.

Reviewer 2 Report

Revised manuscript concerns an important issue of the mechanism of DON exposure on NF-κB inflammatory injury pathway in IPEC-J2 cells. The manuscript is well organized, studies are well documented. The biggest disadvantages are not precisely written aim and scope of research and  an insufficiently indication of what new authors have studied compared to literature. Moreover  I suggest writing a few sentences of conclusions.

Detailed suggestions for corrections are listed below:

Abstract

- all abbreviations should be explained. Currently only obvious DON is explained is missing NF-κB, PDTC, DAO IPEC-J2 and many others. Please provide at least general names for the reader who is not a specialist in this field.

Introduction

- page 1 line 25 - Fusarium graminearum should be italised (name of mould species)

- page 1 line 30 μg·kg-1 please check the method of specifying the units with authors guaidlaine of Toxins and standardize them throughout the text

- page 1 line 32 – „The widespread over standard of DON makes the investigation even more important”. What investigation are you talking about? Nothing has been done about this before unclear sentence - probably shorthand

- page 2 line 52 word „flora” is not correct better will be „microbiota”

- page 2 line 53 „in vitro” should be italised

- the aim and scope of the research should be more clearly written

Results

- „p” or „P”? it also should be italised

- in my opinion, each figure should be self-explanatory, therefore I suggest individual legends under each of tchem (not one explanation under Fig. 1)

discussion

- please specify what's new in the current research in relation to the literature, reading the discussions you get the impression that all the experiments were previously done

- page 8 line „increasingconcentration” should be „increasing concentration”

Materials and Methods

- Chemical and reagents – please consider the list of materials in the table - it should be clearer than the repeated description of "was / were purchased"

- page 9 line 219 „1×105 cells/mL” rather should be „1×10^5 cells/mL”

- line 231 please provide microscop manufacturer data

Lack of conclussions

- I suggest writing a few sentences of conclusions

Author Response

Point 1: Abstract- all abbreviations should be explained. Currently only obvious DON is explained is missing NF-κB, PDTC, DAO IPEC-J2 and many others. Please provide at least general names for the reader who is not a specialist in this field.

Response 1: Thank you for your valuable suggestions. We have revised the full names of all abbreviations when used for the first time according to the reviewer's suggestion.

Point 2: page 1 line 25 - Fusarium graminearum should be italised (name of mold species)

Response 2: Thank you for your careful review. We have changed the name to italics form.

Point 3: page 1 line 30 μg·kg-1 please checks the method of specifying the units with authors guideline of Toxins and standardize them throughout the text

Response 3: Thank you for your valuable suggestions. We have checked and revised the units in the entire manuscript.

Point 4: page 1 line 32 – „The widespread over standard of DON makes the investigation even more important”. What investigation are you talking about? Nothing has been done about this before unclear sentence - probably shorthand

Response 4: Thank you for your careful review. We have revised this sentences “Therefore, further research on DON becomes even more important.” according to reviewer's review.

Point 5: page 2 line 52 word „flora” is not correct better will be „microbiota”

Response 5: Thank you for your careful suggestion. We have changed the word flora following the reviewer's suggestion.

Point 6: page 2 line 53 „in vitro” should be italicized

Response 6: Thank you for your careful review. We have written in vitro in italics form in our revised manuscript.

Point 7: the aim and scope of the research should be more clearly written

Response 7: Thank you for your valuable and important review. According to your valuable suggestions, we have emphasized the aims of this experiment in the introduction of our manuscript.

Point 8:„p” or „P”? it also should be italised

Response 8: Thank you for your valuable review. We have checked and revised these questions all over the manuscript.

Point 9: in my opinion, each figure should be self-explanatory; therefore I suggest individual legends under each of them (not one explanation under Fig. 1)

Response 9: Thank you for your valuable and important suggestion. We have added an individual explanatory under each figure briefly.

Point 10: please specify what's new in the current research in relation to the literature, reading the discussions you get the impression that all the experiments were previously done

Response 10: Thank you for your important review. In this study, IPEC-J2 cells were used as the model. It was found that the inflammatory injury of DON on IPEC-J2 cells was through NF-κB pathway, which was not verified by previous description.

Point 11: page 8 line „increasing concentration” should be „increasing concentration”

Response 11: Thank you for your careful review. We have added spaces between increasing and concentration.

Point 12: Chemical and reagents – please consider the list of materials in the table - it should be clearer than the repeated description of "was / were purchased"

Response 12: Thank you for your careful and valuable comment. We have rechecked and clarified the chemical and reagents used in this experiment.

Point 13: page 9 line 219 „1×105 cells/mL” rather should be „1×10^5 cells/mL”

Response 13: Thank you for your valuable review. We have revised it to 1×105 cells/mL.

Point 14: line 231 please provide microscop manufacturer data

Response 14: Thank you for your careful review. The microscope was provided by Chongqing Chongguang Industry Co., Ltd.

Point 15: I suggest writing a few sentences of conclusions

Response 15: Thank you very much for your valuable suggestion. We have summarized several sentences to illustrate the conclusion of this experiment.

Reviewer 3 Report

The paper shows intersting results based on relevant methods. However, the results have to be written more clearly with more explanations, the discussion need extention and the language needs improvement. All abbriviations have to be written with full names when first mentioned. 

Introduction:

Lines 29-32: I do not understand these sentences.

Lines 62-65: Improve the aims of the study.

Results:

This chapter is nicely and clearly divided in subheadings but nevertheless, some of the text is messy. Furthermore the figures need more explanation. It seems that PDTC may have effect even without DON exposure: this has to be mentioned and discussed. 

The specific comments here are just examples:

Line 83: All cells show irregular shape!

Lines 113-114: Not correct concerning IL-6.

Lines 134-135: Only some of the gene expressions were down-regulated.

Lines 142-145: Messily written

Discussion:

Line 164: Which mycotoxins?

Lines 165-167: The denomination of DON concentrations is different and not easily comparable.   

Lines 177-179: Bring in references that show this permeability effect in vivo.

Line 187: Explain and discuss this pathway

A separate Conclusion chapter is wanted.  

Author Response

Point 1: Lines 29-32: I do not understand these sentences.

Response 1: Thank you very much for your comment. We have revised the suggested sentences to “From 2016 to 2017, 827 complete pigs feed samples and 724 feed components were selected from different provinces of China for testing. It was found that the terminal standard rate of DON content was more than 74.5%, and the average concentration was 450.0-4381.5 μg·kg-1 ” in our revised manuscript.

Point 2: Lines 62-65: Improve the aims of the study.

Response 2: Thank you for your valuable and important review. According to your valuable suggestions, we have emphasized the aims of this experiment in the last paragraph of introduction of the manuscript.

Point 3: Line 83: All cells show irregular shape!

Response 3: Thank you for your careful review. We have revised it according to the reviewer's suggestion.

Point 4: Lines 113-114: Not correct concerning IL-6

Response 4: Thank you for your important comment. We have revised to “concentration” following reviewer's comment.

Point 5: Lines 134-135: Only some of the gene expressions were down-regulated.

Response 5: Thank you for your careful and valuable review. We are sorry that we didn't use precise words. We have improved our manuscript accordingly.

Point 6: Lines 142-145: Messily written

Response 6: Thank you for your comment. We have made changes after confirming the content.

Point 7: Line 164: Which mycotoxins?

Response 7: Thank you for your careful review. There are many of mycotoxins, such as zearalenone, deoxymycoenol and so on all can reduce the survival rate of IPEC-J2 cells.

Point 8: Lines 165-167: The denomination of DON concentrations is different and not easily comparable.

Response 8: Thank you for your valuable review. We have modified the unit of DON concentration in the manuscript to be the same as the experimental content according to reviewer's comment.

Point 9: Lines 177-179: Bring in references that show this permeability effect in vivo.

Response 9: Thank you for your valuable suggestion. We have increased the reference to support the in vivo effect.

Point 10: Line 187: Explain and discuss this pathway

Response 10: Thank you for your valuable and important comment. We have added several sentences about NF-κB signaling pathway in the discussion of the revised manuscript according to reviewer's comment.

Point 11: A separate Conclusion chapter is wanted. 

Response 11: Thank you for your valuable suggestion and comment. We have summarized several sentences to illustrate the conclusion of this experiment. In our revised manuscript we also have included a separate chapter for conclusion.

Round 2

Reviewer 2 Report

The authors have made changes that significantly improve the quality of the article. I believe that in this form it is worth publishing in Toxins